# Attitudes towards Trans Men and Women in Spain: An Adaptation of the ATTMW Scale

**DOI:** 10.3390/ijerph20031872

**Published:** 2023-01-19

**Authors:** Miguel Ángel López-Sáez, Ariadna Angulo-Brunet, Lucas R. Platero, Vincenzo Bochicchio, Oscar Lecuona

**Affiliations:** 1Department of Psychology, Rey Juan Carlos University, 28922 Alcorcón, Spain; 2Faculty of Psychology and Education Sciences, Universitat Oberta de Catalunya, 08018 Barcelona, Spain; 3Department of Humanistic Studies, University of Calabria, 87036 Arcavacata, Italy

**Keywords:** transphobia, transnegativity, homonegativity, sexism, LGBT, trans

## Abstract

This article presents the results of the adaptation and validation of the Attitudes Toward Trans Men and Women (ATTMW) scale—a measure capable of detecting transphobic positions towards trans men and women—in the Spanish context. A total of 310 prospective teachers from different stages of education participated in the study on its adaptation. In order to provide quantitative evidence of validity, confirmatory factor analysis and regression analysis with different constructs and sociodemographic variables were carried out. Internal consistency reliability was adequate. The study demonstrated that the ATTMW is a psychometrically sound instrument for the assessment of attitudes towards trans people, especially with items that categorize trans women and men as second-class people.

## 1. Introduction

### 1.1. Transphobic Drifts in Spain

Although it has been 15 years since the finalized Yogyakarta Principles on LGBT rights was launched as a global charter, some rights still seem to be under debate, even though the data have uncovered a disturbing reality and continue to reveal a social challenge in many countries [1]. The reality is that between January 2008 and September 2022 at least 4696 trans people were murdered worldwide, according to statistics from the Trans Murder Monitoring project [2]. Of these, 327 were killed in 2022, and the vast majority—95%—were trans women [2]. Even though the rights of trans people are still being discussed, in the name of a supposed freedom of expression, many challenges remain. What is most surprising is that this discussion is being accommodated in countries such as Spain, where the rights of LGBT people appeared to have been well established, as reflected by the fact that the country ranked second in the annual Rainbow Europe Index in 2011 (although it has since fallen to eleventh place) [3]. Moreover, the rise of the extreme right and ultraconservative organizations alongside groups of cis women and cis men who exclude trans individuals—better known as Feminism-Appropriating Reactionary Transphobes activists (hereinafter FART)—have had an impact on this reversal, endorsing a single way of being: cisnormative [4]. Moreover, certain academic narratives from Spanish Psychology and Education [5,6] have chosen to align themselves with FART discourses around the defense of sex as an innate biological marker, the impossibility of gender self-determination, the supposed dictatorship of queer ideology, the danger of erasing women, and the silencing of true feminism [4]. To add realism to their narrative, they make use of invented or conveniently reinterpreted cases [4], in addition to taking advantage of their academic positions, which confers a supposed degree of expertise and scientism upon them. If, in the best of cases, they accept the possibility of the existence of trans individuals, it is subject to their oversight [7].

From these positions, they have been able to establish the importance of a phony debate that tries to challenge the scientific consensus of affirmative psychology and international guidelines [8,9], under the premise that they were put in place by a neoliberal queer inquisitive ideology that has irreversible consequences for minors [6]. Accordingly, these groups try to use a language similar to activist languages to present a public social image of an attempt at dialogue and a legitimate claim for their anti-rights positions, but always devaluing and negating the identities of those on the other side of the debate. They construct a polarity between “the trans radical” and their freedom of expression, which is all about safeguarding women and children.

In this respect, FART groups in Psychology and Education have expressed alarm about the profusion of minors visiting gender transition specialists due to—in their words—the visibility of trans issues in the media and social networks, the reduced stigma and the depathologization in addition to the affirmative focus that foments mutilation, in their opinion [6]. Moreover, according to these groups, this entire breeding ground creates a social contagion that produces an outbreak of sudden gender dysphoria, based on the revised study by Litman [10], that ends up in an 80% detransition rate [6]. However, although terms such as “rapid-onset gender dysphoria” have been rejected by much of the scientific literature [9,11,12,13]; although the increase in the number of minors seeking assistance from specialists is due to insufficient or non-existent service in the past [14]; although the role of affirmative therapy is to advocate for general well-being [15] and not to recklessly prescribe single, homogenous reassignment solutions [8]; although the detransition rate is far below 80% [16], ranging between 1% and 2% [17,18] and is usually due to a non-binary tendency or insufficient support [19]; and although being trans is not a contagious fashion (it would be contradictory for one of the most stigmatized and harassed groups [20] to be in fashion), they continue to repeat it, like a mantra.

### 1.2. Keys to Transphobic Narratives

The use of these strategies that advocate a supposedly scientific position disguises—behind a series of displays—the biopolitics of the cisnormative system. The aim of this conservative resistance is to establish an adverse social climate in order to continue structural oppressions such that they contribute to the necropolitics that invisibilize trans beings, while reifying cisnormativity as a state of normalcy. These narratives are based on a cisgenderist cultural identity that perpetuates the belief that only cisgender or cissexual identities are truly real, establishing that identities cannot be shaped according to a compendium of bio-sociocultural factors, since accepting that thesis would be tantamount to choosing an arbitrary identity at will. As a consequence, and in the middle of these anti-trans criticisms, is the issue of the self-determination of gender identity, which is presented as a neoliberal option, as if people could choose as they pleased. In point of fact, self-determination is a legal mechanism to recognize trans identities [21] and a social mechanism that collectivizes the battle against cisnormative violence and responds “to the different degrees of harm people are forced to inhabit” [22] (p. 91).

Therefore, these anti-trans discourses designate a hierarchy that includes a whole range of “rigid beliefs and rules about many aspects of gender, including gender identity, expression, and roles” [23]. These narratives establish a type of monosemic biologization of the categories “man” and “woman,” as if these categories were exempt from any cultural artifice and were a natural fact free from social influences; in other words, as if they were only possible from a cis-genital-centric perspective. People who do not neatly adjust to the rules, appearance, genitals, and phenotype typical of men or women are seen as “sexistly stereotyped and fake,” “disturbed,” and “threatening.”

According to these narratives, trans people are considered “fake,” because any possibility of gender syncretism is eliminated and life is limited to a binary plane where any divergence is a pathological phenotypic expression. Moreover, “passing” and gender expression are monitored, while trans individuals are accused of adopting the sexist stereotypes that are, in turn, the drivers of the transition. They are considered “disturbed,” because they try to establish a regulation that cisnormativizes and eliminates phenotypic diversity according to the argument that people are going through a phase or are mentally unhealthy because they insist on being something that they are not. They are considered “threatening,” because of their desire to slip into a different group, especially in the case of trans women, who are accused of wanting to erase “real women.”

This trio of judgements is aimed at limiting the human rights of trans people—especially their autonomy to decide—to deny and challenge their identity or impose surveillance through evaluation and diagnosis that allows the disorder and, in the best of cases, prescribes the same hormone treatment and surgery for all [24], which may be delayed in the case of trans children. The trans identities that are recognized from anti-trans points of view are required to accept surgery and/or hormones, under the assumption that these are not entirely standard men and women [25]. The demanded standard is an ideal that that can never be reached and serves, in turn, to mark meticulously differentiated distances between trans and cis people, and even between trans individuals themselves. Thus, archaic and rejected presuppositions are once again mobilized [26], establishing that true trans people must want to submit to bodily and genital changes, since they are in the “wrong body.” Yet when they agree to these procedures, trans men and women are scrutinized from the perspective of “normophilia” [27] and considered less authentic than cis individuals. They find themselves on the receiving end of countless aggressions, ranging from the incorrect use of their pronouns and names to the denial of their identity [25,28]. This violent surveillance is justified by a type of “gender-sex right” [29] to validate what anti-trans positions view as a “trick” or “simulation” of a true self.

### 1.3. Conceptualization and Measures of the Construct of Transphobia

The trio of judgements presented above underlie the primary beliefs that form the basis of a range of prejudicial attitudes towards trans people in Spain. Negative attitudes towards trans individuals are termed “transphobia,” and although the term initially referred to emotional repugnance [30], it was later expanded to include a whole range of social and cultural cognitions that accompany emotions and conducts [31,32]. In this respect, some of the current literature has begun to use the term “cisgenderism,” taking the focus from the people at the receiving end of the hatred and putting it on the people who generate it [33].

The first measures that began to conceptualize the construct of transphobia were developed more than a half century ago, although it was not until the beginning of the twenty-first century that special emphasis was placed on developing measures with a certain degree of psychometric soundness to cover more subtle aspects [30,34].

Currently, some studies include up to 83 scales that focus on examining attitudes towards trans people [35], and of them, according to Billard [36], four stand out for being the most validated. However, in Spain only two have been validated: the Genderism and Transphobia Scale and the Trans Attitudes and Beliefs Scale.

The first scale was adapted to the Spanish language by Carrera-Fernández et al. [37], who also reduced Hill and Willoughby’s original scale to 12 items [30]. However, this Spanish validation raised some concerns about the sensitivity to measure more subtle attitudes, since it does not remedy any of the limitations in the earlier version and raises questions about its factorial structure [38]. Additionally, methodological weaknesses have arisen, since the exploratory and confirmatory analyses were done with the same sample, and the sample used for validation only contained adolescents, which could produce generational biases.

The second scale was validated and adapted to Spanish by López-Sáez et al. [38], and although the sensitivity was improved with respect to religious attitudes and civil rights debates, it also raised questions about the internal consistency reliability for one of its dimensions, which might have resulted from the lack of variability in the sample.

On the other hand, some authors have criticized the scales for proposing measures that are too broad and address not only trans people, but also sex/gender dissident and non-binary people [39]. As these realities can generate other types of attitudes, some authors have proposed new measures, for example Billard [36], whose Attitudes Toward Trans Men and Women (ATTMW) scale specifically explores negative attitudes towards trans men and women. In developing the scale, Billard generated 200 items through a qualitative thematic analysis of responses to open-ended questions with a selection of 120 American cis adults representing left- and right-wing political orientations, which guaranteed a range of perceptions. The open-ended questions address different issues related to trans people: associations with the term, the definition and etiology, stereotypes, feelings, and opinions about access to civil rights. Later the 200 items were administered to another sample of 238 adults and the responses were subjected to an Exploratory Factor Analysis (EFA) that revealed two different subscales with 12 items each, 12 for trans women (ATTW; α = 0.99), and 12 for trans men (ATTM; α = 0.97). Finally, the new instrument was administered and a confirmatory factor analysis (CFA) was performed that confirmed the unifactorial structure of each subscale. Again, the analyses confirmed a high internal consistency (α = 0.98 for ATTW; α = 0.94 for ATTM). It is important to note that such high internal consistency reliability values can also be a disadvantage. The instrument also found high correlations with other measures of attitudes towards trans people, homonegativity, sexism, and gender roles. The validity evidence based on the relationship to other variables found a high correlation with other trans measures, but lower with the other measures. The instrument was also a good predictor of conservative and anti-trans politics. Finally, the results found higher negative attitudes among men.

Recent studies have corroborated and expanded the type of connection that transphobia shares with other LGBphobias [38,40,41], sexism [38,40], right-wing political affiliation [41,42,43,44], religiosity [34,45,46], hegemonic or dominant social orientation [32,40,43,47,48], being a cis man [40,41,44], being heterosexual [38,40], and/or having contact with trans people [38,42,49].

These findings are not surprising, since transgenderism challenges the cisheteronormative matrix by breaking with the logic that gender identity, gender expression, and orientation towards desire are permanently determined by the identity assigned at birth [50,51]. Therefore, transphobia shares roots with sexism and gender roles. Sexism is based on the “set of beliefs about the rules, characteristics, behaviors, etc., that are considered appropriate for men and women, as well as beliefs about the relationships that (…) they should have with each other” [52] (p. 274). Moreover, sexist attitudes are not neutral, but seek to maintain the status quo of what has been categorized as “normal”, in other words, of men and cisheteronormativity [53]. In fact, Herek [54], followed by other authors, gathered what different LGBT social movements have highlighted, such as the surveillance of normative femininities and masculinities at the root of discrimination [55]. Therefore, the measurement of attitudes towards trans people is usually related to attitudes towards dissident sexualities [41,44]. In this respect, Herek [54] and other authors use the term “heterosexism” [56,57] and have created different scales to explore these attitudes that denigrate sexual dissidences—and how they break the matrix—and privilege cisheterosexuality.

### 1.4. Current Study

Billard’s proposal [36] is interesting in the current Spanish context, where even though anti-trans positions themselves recognize certain trans identities, they do not consider them to be real men and women, or they view them as second-class people. Moreover, it improves the limitations discussed earlier related to validity and reliability found in other scales [36]. In view of the above, this article adapts and obtains psychometric validity evidence for ATTMW scores. It also provides an approach to the attitudes held by future teachers. Nonetheless, this is an exploratory work, as this type of study is a rarity on the Spanish scene.

## 2. Materials and Methods

### 2.1. Participants

A total of 310 people living in the Region of Madrid participated in this study. Of these, 71.0% were cis women (*n* = 220) and the rest were cis men. No one reported another gender identity. The mean age was 24.6 years (*SD* = 9.4, range = 19 to 80, *Mdn* = 21). Of the total, 83.9% were undergraduate students and 16.1% were postgraduate students. A total of 222 participants (71.6%) reported not having any religion. Of the religious individuals, 88.64% identified as Catholic, 4.55% as agnostic, 3.41% as Protestant, 2.27% as Islamic, and 1.14% as Orthodox.

### 2.2. Procedure

The convenience sample was taken during the 2021–22 academic year through QR codes placed in different student buildings in Primary Education, Children’s Education and the Master’s degree in Teacher Training at the public King Juan Carlos University. We chose this sample in order to study how the rise of FART discourses impacts on people that is studying education and psychology—as indicated in the introduction. The participants used a QR code to access an online test that took approximately 20 min to complete. All of the participants were voluntary and informed of the confidentiality and anonymous nature of their responses. At the beginning of the procedure, participants read the following definitions—similar to Billard’s [36], but based on those established by Perez-Arche and Miller [48]: (a) Trans women: people who were assigned as men at birth, but who identify and live as women and may or may not alter their body through surgical and hormonal interventions to make their expression resemble that of a woman; (b) Trans men: people who were assigned as women at birth, but who identify and live as men and may or may not alter their body through surgical and hormonal interventions to make their expression resemble that of a man.

To adapt the scale, a translation was first done by an academic translator who specializes in gender. For the second step, 12 experts in Gender Psychology—members of different professional and academic task forces with extensive experience in the field—reviewed the adaptation and then performed a double reverse translation to guarantee a similar semantic meaning to the original scale. Then, the set of items was reviewed by three experts in Psychometrics, and a pilot group of five students from Education and five people from a trans association evaluated the clarity and possible redundancy of the items. The procedure behind the study, which forms part of a larger European project, was approved by the University of Calabria Ethics Committee in accordance with the considerations in the Declaration of Helsinki.

### 2.3. Measures

#### 2.3.1. Questionnaire including Sociodemographic Aspects

Participants reported their gender identity (0 = cisgender man; 1 = cisgender woman; [other options given were not selected]), sexual orientation (1 = heterosexual; 2 = bisexual; 3 = gay and lesbian; 4 = open response option [the open response option was not selected]; recoded as 0 = heterosexual; 1 = LGB), age, and academic level (1 = undergraduate; 2 = masters).

The religious items asked about the participants’ self-perception as religious or not (0 = no; 1 = yes).

The contact variables asked about the likelihood that the participants had LGB and T friends or family (0 = no; 1 = yes).

For political affiliation, a single item was used, based on a 5-point Likert scale (1 = right to 5 = left). This was recoded as right (1, 2) and left (3, 4, 5), due to the specific characteristics of the Spanish population, which have been observed in other studies [53].

#### 2.3.2. Attitudes toward Trans Men and Women (ATTMW)

We adapted the measure developed by Billard [36] to Spanish. This measure has two subscales: attitudes toward trans men (ATTM) and attitudes toward trans women (ATTW). Each of the unidimensional scales has 12 items and we used a 7-point scale (1 = strongly disagree, 7 = strongly agree). For the ATTM scale all items were direct except for item 6 (“Trans men seem absolutely normal to me”). For the trans woman scale all items were direct. A higher score in both scales indicates a favorable attitude toward transgenderism. The psychometric properties of this scale are described in detail in the Results section, as they are one of the focal points of this work.

#### 2.3.3. Ambivalent Sexism Inventory (ASI)

We used a modified version of ASI that does not have reversed items [58], validated for the Spanish population by Expósito [59]. This is a 22-item scale and uses a 6-point Likert scale (0 = strongly disagree; 5 = strongly agree). In this sample we obtained adequate goodness of fit indexes (GOFI) for a unidimensional scale (𝜒^2^ = 698.3, *CFI* = 0.99, *TLI* = 0.97, *RMSEA* [90%CI] = 0.09 [0.08, 0.09]) and excellent internal consistency reliability for the sum scores (ω = 0.95, α = 0.95). Evidence for a two-factor correlated model was also found (𝜒^2^ = 35,893.9, *CFI* = 0.99, *TLI* = 0.99, RMSEA [90%CI] = 0.07 [0.07, 0.08]). Both factors had good internal consistency: benevolent (ASI-BS; ω = 0.88, α = 0.88) and hostile (ASI-HS; ω = 0.93, α = 0.93).

#### 2.3.4. Multidimensional Heterosexism Inventory (MHI)

We used the 23-item scale for measuring heterosexism [57]. The scale uses a 7-point Likert scale (1 = strongly disagree; 7 = strongly agree), and four dimensions: paternalistic heterosexism (MHI-PH), aversive heterosexism (MHI-AVH), amnestic heterosexism (MHI-AMS), and positive stereotypic heterosexism (MHI-PSH). Higher scores represent higher heterosexism, respectively. In this sample, we obtained an excellent GOFI for a 4-factor correlated model (𝜒^2^ = 24,854.1, *CFI* = 1.00, *TLI* = 1.00, *RMSEA* [90%CI] = 0.02 [0.00, 0.04]). The internal consistency was adequate for all four subscales: paternalistic heterosexism (ω = 0.95, α = 0.94), aversive heterosexism (ω = 0.92, α = 0.92), amnestic heterosexism (ω = 0.83, α = 0.88), and positive stereotypic heterosexism (ω = 0.86, α = 0.87).

#### 2.3.5. Social Dominance Orientation-Short Form (SDO)

This is an eight-item 7-point scale (1 = strongly oppose, 7 = strongly favor) measuring orientation towards social domination [60] that was previously validated for the Spanish population [61]. A higher score indicates higher social dominance (e.g., “some groups of people are simply inferior to other groups”). This scale has four inverse items that were reverse-coded prior to data analysis. In this sample we found positive evidence for an essentially unidimensional scale using a method factor (𝜒^2^ = 1283.6, CFI = 0.97, TLI = 0.96, RMSEA [90%CI] = 0.08 [0.05, 0.11]). Despite good internal structure validity evidence, the internal consistency reliability was not adequate (ω = 0.64, α = 0.57).

### 2.4. Data Analysis

For the ATTMW test, descriptive statistics and frequency tables were created to see the item distribution. The power analysis for the psychometric analyses reported adequate properties for this sample size applied to the ATTMW with our data [62].

To assess validity evidence related to the internal structure of the ATTMW and external variables we performed CFA models and assessed the internal consistency reliability following the guidelines of Doval et al. [63]. For that reason, we implemented the unweighted least squares estimator for CFA and provide omega and Cronbach’s alpha reliability coefficients for all the measures. Regarding the GOFI, we followed the criteria provided by Hu and Bentler [64] for considering adequate fit: CFI and TLI greater than 0.90, and RMSEA smaller than 0.08. We first tested two models based on Billard’s work. In light of the lack of discrimination found in the results, we also tested alternative models.

In order to assess the relationship with other variables, we followed two strategies. The first step included a bivariate analysis and descriptive statistics for all the variables and the total ATTM score and total ATTW score. The second step included two multiple linear regression models, using ATTM and ATTW as dependent variables, and introduced the relevant variables as independent variables.

## 3. Results

To provide evidence of the psychometric quality of the ATTMW, first the response distributions of the response items were examined. Second, evidence of validity related to the internal structure was studied; and finally, evidence related to other variables was also considered.

### 3.1. Item Descriptive Statistics and Confirmatory Factor Analysis

Table 1 shows the descriptive statistics for the transphobia scale for trans men and trans women. As can be seen from the extreme values of the mean, as well as the kurtosis values (between 5.7 and 23.0), there is a marked floor effect in all the items (and a ceiling in the reverse item, attm06).

Therefore, although the item distribution was apparently continuous, we considered examining the response distribution by frequencies (see Figure 1). As can be seen, except for attm06, more than 70% of the participants had very extreme responses. In this case, this could be interpreted as the people in the sample showing non-transphobic attitudes towards trans men and women. In the case of the item “Trans men seem absolutely normal to me”, there is more variability in the responses.

Following Billard’s study, we examined two different models, one for ATTM and another for ATTW. The standardized factor loadings for ATTM can be found in Table 1. As can be seen, all factor loadings are high except for item attm06. This one-factor model yielded adequate GOFI *(*𝜒^2^ = 106.5, *df* = 54, *CFI* = 0.99, *TLI* = 0.99, RMSEA [90%CI] = 0.06 [0.04, 0.07]). The internal consistency of the sum scores was also adequate (ω = 0.889, α = 0.887).

Similar results were obtained for the ATTW (Table 2). In this case, all the factor loadings were high and positive and the GOFI were excellent for the one-factor CFA (𝜒^2^ = 99.2, *df* = 54, *CFI* = 0.99, *TLI* = 0.99, *RMSEA* [90%CI] = 0.05 [0.04, 0.07]). The internal consistency reliability was excellent (ω = 0.931, α = 0.931).

We also tested a two-factor correlated model with the ATTM and ATTW scales. This model had adequate GOFI (𝜒^2^ = 765.11, *df* = 251, *CFI* = 0.98, *TLI* = 0.98, *RMSEA* [90%CI] = 0.08 [0.04, 0.09]). In this case, the correlation between each factor was 0.98, indicating a lack of discrimination between the two scales. Based on this model, we obtained the same values for the omega internal consistency reliability. We then tested a one-factor model and obtained similar GOFI (𝜒^2^ = 767.93, *df* = 252, *CFI* = 0.98, *TLI* = 0.98, *RMSEA* [90%CI] = 0.08 [0.08, 0.09]) and excellent GOFI (ω = 0.952, α = 0.952). Finally, we attempted a hierarchical second-order factor and two first-order factors, but this was unavailable due to estimation issues.

Taking into account the results obtained in the different models studied, we considered the model that most closely matched the one proposed by Billard, with two separate scales of transphobia towards men and women.

### 3.2. Relationship with Other Variables

First, we assessed whether there were differences in the two total scores considering the sociodemographic factors as shown in Table 3. We found that the means of the ATTM and ATTW were similar (1.46 and 1.39, respectively); that is, the participants perceived low transphobia towards trans men and trans woman. On the other hand, cis men had greater transphobia than cis women, heterosexuals greater transphobia than LGB, and religious people showed higher transphobia than non-religious people (who directed their transphobia toward men).

Regarding contact with LGB individuals, there were no differences in the levels of transphobia for people who had LGB family or friends. However, there were differences between the people with trans relatives and people without. Those who had no trans friends had greater transphobia that those who did not. Regarding friendship, similar results were found; having LGB friends was not associated with different scores for transphobia levels.

Table 4 presents the results of analyzing the relationship between the total score of the two transphobia scales and the relevant external variables. Regarding SDO, the scores are generally below the midpoint of the scale, indicating a potentially low social dominance. There are also extreme low scores in ASI-BS and ASI-HS and in all the subscales of the MHI.

The correlations related to transphobia towards men are moderate to high with all the variables (0.36 to 0.75). The high correlation with MHI-AVH and MHI-PSH is notable. Regarding transphobia towards trans women, there are also moderate-to-high correlations (0.29 to 0.69). The strong correlation between aversive and stereotypic heterosexism is important.

To evaluate whether the external variables used in the study were predictors of ATTM and ATTW, we used multiple linear regression. Table 5 shows the models generated for ATTM and Table 6 shows the models generated for ATTW. In a first step, all the study variables were introduced (Step 1) and in the following steps, those not statistically significant or not contributing to the change of r2 in the models were eliminated. The results of the final model (Step 2) are discussed below.

In this study, the most important variable predicting the ATTM score was MHI-AVH. For each point on the MHI-AVH, the ATTM score increased by 0.51 points (95%CI 0.45–0.57, *p* < 0.001). On the other hand, consistent with the results of the bivariate analyses, we found that religious people have a slightly higher score of 0.16 points (95%CI 0.03–0.29, *p* = 0.02) than non-religious people for ATTM. For each point increase on the SDO scale, there was a 0.02 (95%CI 0.01–0.03, *p* < 0.001) increase in ATTM. This model explains the 58% variability in the ATTM scores.

Regarding the ATTW scale, it was also possible to predict the 50% variability of these scores. In this case, as with ATTM, the most predictive variable was MHI-AVH, where for each point increase in MHI-AVH there was a 0.45 increase in ATTW (95%CI 0.37–0.53, *p* < 0.001). In this case, for each point increase in SDO there was an increase of 0.02 in ATTW (95%CI 0.00–0.03, *p* < 0.001). We also found an effect with MHI-PSH; for each point increase in heterosexism there was a 0.12 increase in ATTW.

## 4. Discussion

The main goal of this study was to adapt the ATTMW to the Spanish context with good psychometric soundness.

In this respect, the CFA corroborated that the two subscales (ATTM, ATTW) both have a good overall unidimensional adjustment. As with the original scale, a lower factor load repeated with attm06: “Trans men seem absolutely normal to me.” This value and the higher response variability for this item could explain the lower acceptance of trans men regarding the subjectivity surrounding what is considered normal. Normalcy might refer to the criteria used to determine gender identity, and some people do use criteria more linked to phenotype and genitality, while others use more sociocultural criteria [65,66]. Additionally, the resulting internal consistency coefficients suggest an adequate reliability for ATTM and an excellent reliability for ATTW.

With regard to the evidence of discrimination between the two scales, the correlation is very high. This is an expected result given that both scales measure closely related concepts. On the other hand, regarding the relations with other constructs with which they share roots, the ATTM correlations are usually higher, in line with the findings of Billard [36]. As seen, alternative models were considered, with the intention of shedding light on the problems of discrimination in both scales. The GOFIs in the one-dimensional models were adequate. In contrast, when examining all the items as a single scale (either unidimensional or two-factor correlated), although the fit was good, the upper range of the RMSEA was outside the recommended range. This could be explained, in part, by a method effect. More concretely, items from both subscales partially share content. Thus, the tested internal structures were appropriate in this sample, and we chose the originally proposed model by Billiard. Since our aim was to study the psychometric properties of the ATTMW in the Spanish population, theoretical analyses related to a specific structure go beyond the scope of this study. Future studies could propose and discuss the suitability of modifying or reducing these scales in the Spanish population or other sample groups.

The highest correlations in both subscales occurred with positive stereotypic heterosexism (MHI-PSH) and aversive heterosexism (MHI-AVH), which may be related to the fact that the semantic composition of the items could refer to a stereotypical concept of what it means to be a man and woman, as occurs with MHI-PSH. Additionally, the ways of indicating rejection are less explicit and use “natural” arguments, e.g., MHI-AVH [57]. In this vein, the findings from the regression analysis corroborate how these attitudes that ignore the implications of sexual dissidence for leading a livable life—MHI-A—predict greater transphobia towards trans men and women.

The regression analyses also show that SDO has some predictive potential for both subscales, supporting what has been found in other studies [40,48]. Social dominance attitudes back hierarchical structures among individuals with a high status quo, and supporting transgenderism means rejecting these hegemonic points of view.

To a lesser extent, religiosity is a predictive variable for attitudes towards trans men, which is congruent with the binary, stereotypical conviction about the world offered by Christianity and, particularly, Catholicism. This predictor has also been highlighted in recent reviews, such as Campbell et al. [45] and studies by Kanamori and Xu [67]. The fact that religiosity, primarily Catholicism, predicts transphobia towards trans men but not so much towards trans women may be related to a more pronounced rejection of those who want to change from being, metaphorically, “Adam’s rib” to being the subject that created them, like Adam himself.

Religious individuals also have higher transphobia, as found in other studies [38,46]. As also found in other studies [38,40], cis men show higher transphobia than cis women, as do heterosexuals and people with right-wing political orientations. Having LGB friends or family members does not make a difference in being more or less transphobic, but having trans family members does. Having trans friends is especially related to lower transphobic attitudes towards men, a finding that has not appeared in earlier studies. These results seem to confirm Allport’s contact hypothesis, as found in other studies [68,69], since social contact with trans individuals lowers transphobic attitudes. However, it is important to stress the importance of the finding that this familiarity occurs with trans and not just LGB individuals, in order to foster this relationship of allies. In this respect, one important limitation in the methodology is that only nine participants had trans family members, while 75 participants reported having trans friends. Although these revelations could be used as possible risk factors, our sample had low levels of transphobia.

## 5. Conclusions

In conclusion, the ATTMW is a significant instrument in the field of measuring negative attitudes towards trans people, specifically towards trans men and women. A culture that does not recognize trans men and women stays cissexist and anti-syncretic, normalizing and prioritizing not only cis experiences, but specific cis experiences that cling to stereotypical models of the cisheterosexual matrix. These attitudes are active manifestations that defend a biologicist and essentialist perspective of what it means to be a man or woman. Moreover, they are expressed “learnedly” by those who feel they have the ability to identify a person’s true gender identity with more validity than the person themself.

This all comes at a critical moment in Spain, when the pressure to stop the advance of trans rights is intense. Therefore, instruments like the one examined here play an essential role in dismantling and making visible transprejudices.

This article also contributes to the study of prejudice towards trans people and the configuration of an emerging field of study within social psychology that focuses on investigating the causes of social discomfort without pathologizing trans people, and putting the problem on the people who hold these prejudiced attitudes [70].

Nonetheless, future work must explore how the ATTMW behaves with more heterogenous and representative samples, since the feminization, secularization, and left-wing political orientation of this sample may have helped produce polarized effects with less variability. In conclusion, it is important to critique the scale itself, which measures attitudes towards transgenderism following the categories of “man” and “woman,” which, significantly, partly excludes non-binary trans individuals.

## Figures and Tables

**Figure 1 ijerph-20-01872-f001:**
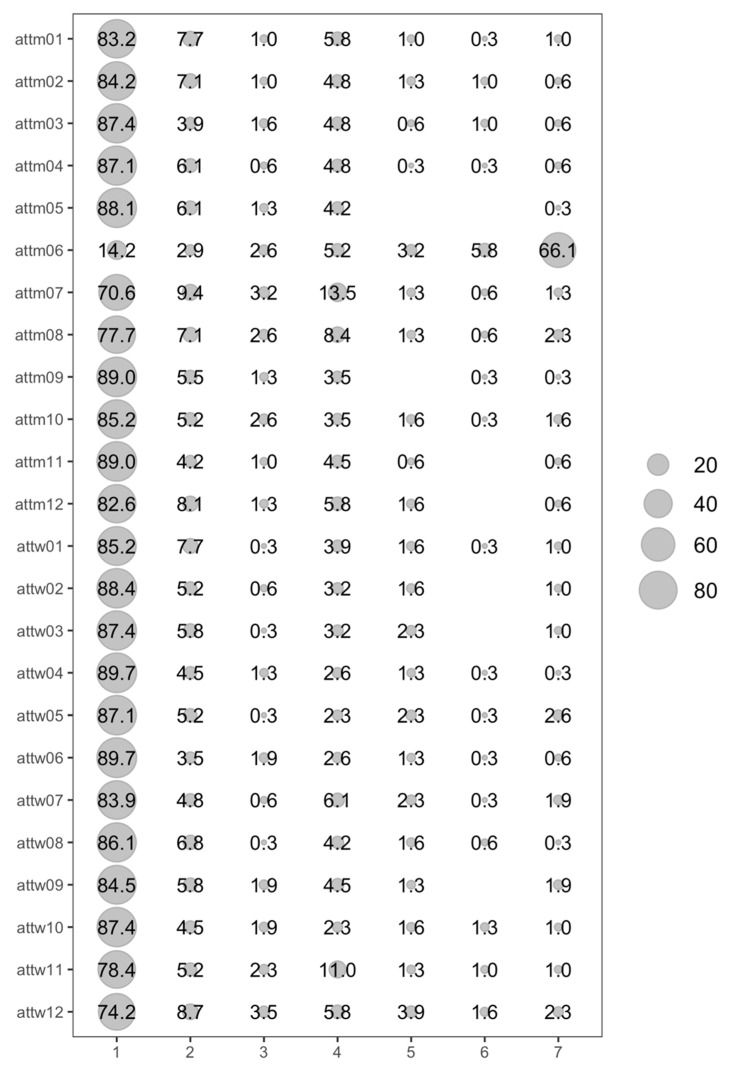
Item Distribution for ATTMW.

**Table 1 ijerph-20-01872-t001:** ATTM Item Content, Descriptive Statistics, and Standardized Factor Loadings.

	M	SD	Min	Max	sk	k	Factor Loadings
attm01-Trans men will never really be men. [Los hombres trans nunca serán realmente hombres.]	1.4	1.0	1	7	3.2	13.5	0.86
attm02-Trans men are not really men. [Los hombres trans no son realmente hombres.]	1.4	1.0	1	7	3.2	13.2	0.87
attm03-Trans men are only able to look like men, but not be men. [Los hombres trans solo pueden parecer hombres, pero no serlo.]	1.3	1.0	1	7	3.4	14.8	0.86
attm04-Trans men are unable to accept who they really are. [Los hombres trans son incapaces de aceptar quiénes son realmente.]	1.3	0.9	1	7	3.7	18.2	0.76
attm05-Trans men are trying to be someone they’re not. [Los hombres trans intentan ser alguien que no son.]	1.2	0.7	1	7	3.9	20.2	0.88
attm06-Trans men seem absolutely normal to me. (R) [Los hombres trans me parecen absolutamente normales.]	5.6	2.2	1	7	−1.3	2.9	−0.21
attm07-Trans men are denying their DNA. [Los hombres trans niegan su ADN.]	1.7	1.3	1	7	1.8	5.7	0.55
attm08-Trans men cannot just “identify” as men. [Los hombres trans no pueden simplemente “identificarse” como hombres.]	1.6	1.3	1	7	2.5	8.6	0.47
attm09-Trans men are misguided. [Los hombres trans están equivocados.]	1.2	0.8	1	7	4.2	23.0	0.83
attm10-Trans men are unnatural. [Los hombres trans no son naturales.]	1.4	1.1	1	7	3.3	14.3	0.77
attm11-Trans men don’t really understand what it means to be a man. [Los hombres trans no entienden realmente lo que significa ser un hombre.]	1.3	0.9	1	7	3.9	19.2	0.90
attm12-Trans men are emotionally unstable. [Los hombres trans son emocionalmente inestables.]	1.4	1.0	1	7	3.0	12.1	0.79

Note. Items followed by (R) should be reverse-scored before calculating totals.

**Table 2 ijerph-20-01872-t002:** ATTW Item Content, Descriptive Statistics, and Standardized Factor Loadings.

	M	SD	Min	Max	sk	k	Factor Loadings
attw01-Trans women will never really be women. [Las mujeres trans nunca serán realmente mujeres.]	1.3	1.0	1	7	3.5	16.0	0.87
attw02-Trans women are only able to look like women, but not be women. [Las mujeres trans sólo pueden parecer mujeres, pero no serlo.]	1.3	0.9	1	7	3.9	19.1	0.90
attw03-Trans women are not really women. [Las mujeres trans no son realmente mujeres.]	1.3	1.0	1	7	3.7	16.9	0.87
attw04-Trans women are trying to be someone they’re not. [Las mujeres trans intentan ser alguien que no son.]	1.2	0.8	1	7	4.1	20.7	0.86
attw05-Trans women are unnatural. [Las mujeres trans no son naturales.]	1.4	1.2	1	7	3.5	14.4	0.66
attw06-Trans women don’t really understand what it means to be a woman. [Las mujeres trans no entienden realmente lo que significa ser una mujer.]	1.3	0.9	1	7	4.0	20.1	0.91
attw07-Trans women cannot just “identify” as women. [Las mujeres trans no pueden simplemente “identificarse” como mujeres.]	1.5	1.2	1	7	2.9	10.7	0.60
attw08-Trans women are unable to accept who they really are. [Las mujeres trans no pueden aceptar quiénes son realmente.]	1.3	0.9	1	7	3.4	14.6	0.84
attw09-Trans women only think they are women. [Las mujeres trans sólo piensan que son mujeres.]	1.4	1.1	1	7	3.3	14.2	0.59
attw10-Trans women are defying nature. [Las mujeres trans desafían a la naturaleza.]	1.3	1.1	1	7	3.6	15.6	0.73
attw11-Trans women are denying their DNA. [Las mujeres trans niegan su ADN.]	1.6	1.3	1	7	2.2	7.0	0.63
attw12-There is something unique about being a woman. that trans women can never experience. [Hay algo único en ser mujer que las mujeres trans nunca podrán experimentar.]	1.7	1.4	1	7	2.2	6.8	0.60

**Table 3 ijerph-20-01872-t003:** Differences between Sociodemographic Variables in ATTM and ATTW.

	ATTM			ATTW		
Variable	M(SD)	Median	Test	M(SD)	Median	Test
General	1.46(0.79)	1.17		1.39(0.83)	1.00	
Gender Identity			t = −3.15 **			t = −2.80 **
- Cis woman (*n* = 220)	1.36(0.68)	1.08		1.29(0.71)	1.00	
- Cis man (*n* = 90)	1.71(0.97)	1.42		1.62(1.02)	1.17	
Sexual orientation			𝜒 = 6.93 **			𝜒^2^ = 1.10 **
- Heterosexual (*n* = 276)	1.50(0.82)	1.17		1.41(0.87)	1.00	
- Lesbian/Gay (*n* = 34)	1.16(0.25)	1.00		1.18(0.30)	1.00	
Religious			t = −3.16 **			−2.28 *
- No (*n* = 222)	1.36(0.65)	1.08		1.31(0.72)	1.00	
- Yes (*n* = 88)	1.73(1.01)	1.46		1.58(1.03)	1.00	
LGB Family			t = 0.05			t = 0.08
- No (*n* = 174)	1.46(0.82)	1.17		1.38(0.83)	1.00	
- Yes (*n* = 136)	1.47(0.76)	1.17		1.39(0.83)	1.00	
Trans Family			𝜒^2^ = 1.07			𝜒^2^ = 0.05
- No (*n* = 301)	1.47(0.8)	1.17		1.39(0.84)	1.18	
- Yes (*n* = 9)	1.15(0.21)	1.08		1.16(0.23)	1.16	
LGB Friends			𝜒^2^ = 0.83			𝜒^2^ = 0.04
- No (*n* = 290)	1.44(0.75)	1.17		1.62(1.20)	1.00	
- Yes (*n* = 20)	1.81(1.15)	1.25		1.37(0.80)	1.00	
Trans Friends			𝜒^2^ = 10.07 **			𝜒^2^ = 3.46 *
- No (*n* = 235)	1.52(0.8)	1.25		1.43(0.84)	1.00	
- Yes (*n* = 75)	1.28(0.71)	1.00		1.25(0.76)	1.00	
Political affiliation			t = 3.67 **			t = 2.99 *
- Left spectrum (*n* = 120)	1.68(0.93)	1.42		1.57(1.00)	1.00	
- Center-right spectrum (*n* = 190)	1.33(0.65)	1.04		1.27(0.67)	1.00	

Note. *: *p* < 0.05. **: *p* < 0.001, ** t = *t*-test, 𝜒^2^ = U-Mann–Whitney test.

**Table 4 ijerph-20-01872-t004:** Descriptive Statistics and Correlation between ATTM and ATTW and External Variables.

	M	SD	Min	Max	(1)	(2)	(3)	(4)	(5)	(6)	(7)	(8)	(9)
(1) A. Towards Trans men (ATTM)	1.5	0.8	1.0	6.5	1.00								
(2) A. Towards Trans women (ATTW)	1.4	0.8	1.0	7.0	0.89	1.00							
(3) Social Dominance (SDO)	15.9	6.5	8.0	40.0	0.36	0.33	1.00						
(4) Benevolent Sexism (ASI-BS)	0.9	0.8	0.0	3.8	0.42	0.36	0.33	1.00					
(5) Hostile Sexism (ASI-HS)	0.9	1.0	0.0	4.2	0.49	0.43	0.40	0.79	1.00				
(6) Paternalistic Heterosexism (MHI-PH)	0.9	1.4	0.0	7.0	0.46	0.42	0.24	0.41	0.37	1.00			
(7) Aversive Heterosexism (MHI-AVH)	1.7	1.1	1.0	7.0	0.75	0.69	0.30	0.52	0.58	0.56	1.00		
(8) Amnesic Heterosexism (MHI-AMH)	1.9	1.2	0.8	7.0	0.45	0.44	0.22	0.35	0.40	0.35	0.51	1.00	
(9) Positive Stereotypic Heterosexism (MHI-PSH)	1.6	0.9	1.0	7.0	0.53	0.52	0.25	0.31	0.37	0.43	0.61	0.40	1.00

Note: All *p*-values < 0.001.

**Table 5 ijerph-20-01872-t005:** Regression Coefficients for ATTM.

Variable	B	95% CI for B		SE	t	*p*	r^2^_Adj_
		LL	UL				
Step 1							0.58
Constant	0.31	0.09	0.53	0.11	2.80	0.01	
Gender Identity	0.04	−0.11	0.19	0.08	0.51	0.61	
Sexual Orientation	−0.08	−0.28	0.11	0.10	−0.85	0.40	
Trans Family	−0.19	−0.53	0.16	0.18	−1.06	0.29	
Trans Friends	−0.04	−0.19	0.10	0.07	−0.62	0.54	
Religious	0.15	0.02	0.28	0.07	2.22	0.03	
Political Affiliation	−0.02	−0.16	0.12	0.07	−0.29	0.77	
Social Dominance (SDO)	0.01	0.00	0.02	0.01	2.85	0.01	
Benevolent Sexism (ASI-BS)	−0.06	−0.18	0.06	0.06	−0.94	0.35	
Hostile Sexism (ASI-HS)	0.03	−0.07	0.14	0.05	0.63	0.53	
Paternalistic Heterosexism (MHI-PH)	0.01	−0.04	0.06	0.03	0.53	0.60	
Aversive Heterosexism (MHI-AVH)	0.43	0.35	0.52	0.04	9.98	<0.001	
Amnesic Heterosexism (MHI-AMH)	0.05	−0.01	0.10	0.03	1.64	0.10	
Positive Stereotypic Heterosexism (MHI-PSH)	0.07	−0.01	0.15	0.04	1.77	0.08	
Step 2							0.58
Constant	0.31	0.15	0.47	0.08	3.84	<0.001	
Religious	0.16	0.03	0.29	0.06	2.44	0.02	
Social Dominance (SDO)	0.02	0.01	0.03	0.00	3.58	<0.001	
Aversive Heterosexism (MHI-AVH)	0.51	0.45	0.57	0.03	17.81	<0.001	

**Table 6 ijerph-20-01872-t006:** Regression Coefficients for ATTW.

Variable	B	95% CI for B		SE	t	*p*	r^2^_Adj_
		LL	UL				
Step 1							0.50
Constant	0.21	−0.05	0.46	0.13	1.60	0.11	
Gender Identity	0.03	−0.14	0.21	0.09	0.37	0.71	
Sexual Orientation	0.01	−0.21	0.23	0.11	0.08	0.93	
Trans Family	−0.13	−0.53	0.27	0.20	−0.63	0.53	
Trans Friends	−0.02	−0.18	0.15	0.08	−0.22	0.82	
Religious	0.07	−0.08	0.23	0.08	0.94	0.35	
Political Affiliation	−0.01	−0.18	0.15	0.08	−0.18	0.85	
Social Dominance (SDO)	0.01	0.00	0.03	0.01	2.48	0.01	
Benevolent Sexism (ASI-BS)	−0.07	−0.21	0.07	0.07	−0.98	0.33	
Hostile Sexism (ASI-HS)	0.02	−0.10	0.15	0.06	0.33	0.74	
Paternalistic Heterosexism (MHI-PH)	0.01	−0.05	0.07	0.03	0.35	0.72	
Aversive Heterosexism (MHI-AVH)	0.42	0.32	0.52	0.05	8.35	<0.001	
Amnesic Heterosexism (MHI-AMH)	0.06	0.00	0.13	0.03	1.86	0.06	
Positive Stereotypic Heterosexism (MHI-PSH)	0.10	0.01	0.19	0.05	2.08	0.04	
Step 2							
Constant	0.22	0.03	0.40	0.10	2.25	0.03	0.50
Social Dominance (SDO)	0.02	0.00	0.03	0.01	2.88	0.01	
Aversive Heterosexism (MHI-AVH)	0.45	0.37	0.53	0.04	11.15	<0.001	
Positive Stereotypic Heterosexism (MHI-PSH)	0.12	0.03	0.21	0.05	2.53	0.01	

## Data Availability

The raw data supporting the conclusions in this article will be made available by the authors, without undue reservation, to any qualified researcher.

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
