# Peer review of "Attitudes towards Trans Men and Women in Spain: An Adaptation of the ATTMW Scale"

_ijerph, 2023, doi:10.3390/ijerph20031872_

Round 1

Reviewer 1 Report

Dear authors, 

This manuscript is highly original and touches on a very pertinent topic. I read it with great interest. I have some suggestions :

1) How were the experts in Gender Psychology defined ?

2) The term  'trans-exclusionary radfem activists' at line 30 and 31 should be title-cased.

3) In the introduction, the authors discuss the dilemma and issues faced by the trans community in Spain. It will be beneficiary to the readers if a global perspective of the LGBTQ community is referenced in the introduction :

Juhari, J. A., Gill, J. S., & Francis, B. (2022, September). Coping Strategies and Mental Disorders among the LGBT+ Community in Malaysia. In Healthcare (Vol. 10, No. 10, p. 1885). MDPI.

Author Response

Thank you very much for the review tips. We have incorporated them all. Below is our response to the two reviewers. The changes have been included in green within the manuscript.

Reviewer 1

This manuscript is highly original and touches on a very pertinent topic. I read it with great interest. I have some suggestions :

1) How were the experts in Gender Psychology defined?

We have clarified the text as follows: “For the second step, 12 experts in Gender Psychology – members of different professional and academic task forces with extensive experience in the field – reviewed the adaptation and then performed a double reverse translation to guarantee a similar semantic meaning to the original scale.”

2) The term 'trans-exclusionary radfem activists' at line 30 and 31 should be title-cased.

We have modified the text.

3) In the introduction, the authors discuss the dilemma and issues faced by the trans community in Spain. It will be beneficiary to the readers if a global perspective of the LGBTQ community.

In line with the recommendation, we have applied this global perspective and then framed it in the Spanish context.

Reviewer 2 Report

I read the article titled ‘Attitudes Towards Trans Men and Women in Spain: An Adaptation of the ATTMW Scale’ carefully and with interest. It presents the results of the adaptation and validation of the ATTMW to the Spanish context. I think than the manuscript has several strengths and it could be of interest to the readers of IJERPH, and I would be glad to recommend it for publication. At the same time, I believe that the manuscript still has some margins of improvements, that the authors might implement through the following minor revisions I suggest:

1. In the introduction, the Annual Europe Index of 2011 gives a not very up-to-date view of the Spanish situation. Perhaps the authors could use the 2022 ILGA report  (https://www.ilga-europe.org/report/rainbow-europe-2022/ )and the 2019 Eurobarometer on the social acceptance of LGBTIQ people in the EU (https://commission.europa.eu/system/files/2019-10/ebs_493_data_fact_lgbti_eu_en-1.pdf ). Also, for a more general consideration about the sexual prejudice and stereotyping in modern societies, the authors might also consider and cite the following reference: Salvati, M., De Cristofaro, V., Fasoli, F., Paolini, D., & Zotti, D. (2020). Introduction to the special issue: Sexual prejudice and stereotyping in modern societies. Psicologia sociale, 15(1), 5-14. DOI: 10.1482/96291 )

2. In the participants section I read no mentions to sample size adequacy. How was the adequate sample size chosen? Di the author run an a priori power analysis/post-hoc sensitivity analysis? Did the authors refer to similar samples in previous studies? I would recommend adding some considerations and reassurance on such an aspect in this section.

3. In the procedure section, I would like to read more info about the sampling choices. When were the data collected, who collected them, how (online/pensil-paper…), duration of the questionnaire... I would also better argue the choice to focus the sample on a particular type of participant, a choice that had an impact on the generalizability of the results, (which should be cited in the limitation sections).

4. I think that the weakest aspect of this tool is the excessive correlation between the two sub-scales (.89). I wonder if it makes sense to keep the two dimensions or if it is not preferable to carry out analyzes to support a mono-dimensional structure. Also, 24-total item for a tool with two dimensions which are so high correlated maybe can be reduced. Perhaps, rather than conducting two separate confirmatory analyzes for the two instruments, it could be useful to do a single CFA on the 24 items to see if the two-factor structure is confirmed... Did the author so any considerations about this? If the authors choose to maintain the two independent tools, maybe additional theoretical justifications for such a choice would be appropriate.

5. I would recommend the authors to deepen the limitations section of the manuscript. The authors only cite the fact that that only nine participants had trans family members. There are other aspects that the authors might consider (the generalizability of the results, the high correlations of the two tools, etc… -some aspects are reported in the conclusions-)

6. I appreciated the conclusions section. I would stress more with a couple of sentences the relevance to represent more the ‘T’ and the ‘Q’ people of the LGBTQ acronym, in the scientific research too. This paper contributes to this, and such a tool might be useful for other researchers who want to do research on LGBTQ+ issues, making trans people less invisible and ignored. Maybe the authors might read and cite the following reference to elaborate more on this point: Salvati, M., & Koc, Y. (2022). Advancing research into the social psychology of sexual orientations and gender identities: Current research and future directions. European Journal of Social Psychology, 52(2), 225-232. https://doi.org/10.1002/ejsp.2875

I hope the authors find my suggestions useful, and I thank the editor for the opportunity to review this work.

Best regards,

Author Response

Thank you very much for the review tips. We have incorporated them all. Below is our response to the two reviewers. The changes have been included in green within the manuscript.

Reviewer 2

  1. In the introduction, the Annual Europe Index of 2011 gives a not very up-to-date view of the Spanish situation. Perhaps the authors could use the 2022 ILGA report  (https://www.ilga-europe.org/report/rainbow-europe-2022/ )and the 2019 Eurobarometer on the social acceptance of LGBTIQ people in the EU (https://commission.europa.eu/system/files/2019-10/ebs_493_data_fact_lgbti_eu_en-1.pdf ). Also, for a more general consideration about the sexual prejudice and stereotyping in modern societies, the authors might also consider and cite the following reference: Salvati, M., De Cristofaro, V., Fasoli, F., Paolini, D., & Zotti, D. (2020). Introduction to the special issue: Sexual prejudice and stereotyping in modern societies. Psicologia sociale15(1), 5-14. DOI: 10.1482/96291 )

We have included these references and clarified the current Spanish situation.

  1. In the participants section I read no mentions to sample size adequacy. How was the adequate sample size chosen? Di the author run an a priori power analysis/post-hoc sensitivity analysis? Did the authors refer to similar samples in previous studies? I would recommend adding some considerations and reassurance on such an aspect in this section.

We thank the reviewer for this comment. We did not run an a priori power analysis, but we conducted a compromise power analysis on the CFAs using sempower (https://sempower.shinyapps.io/sempower/). Using our sample size (310), alpha/beta ratio of 1 (implying 95% power and confidence), our RMSEA (0.06), degrees of freedom (54) and number of manifest variables (12), we obtained an implied power >99%. We interpret this as a good result that should, however, be viewed with caution since it was not selected a priori. We have included this in the “Participants” section and emphasized it in the “Limitations” section.

  1. In the procedure section, I would like to read more info about the sampling choices. When were the data collected, who collected them, how (online/pensil-paper…), duration of the questionnaire... I would also better argue the choice to focus the sample on a particular type of participant, a choice that had an impact on the generalizability of the results, (which should be cited in the limitation sections).

We have included all these details in the “Procedure” section.

  1. I think that the weakest aspect of this tool is the excessive correlation between the two sub-scales (.89). I wonder if it makes sense to keep the two dimensions or if it is not preferable to carry out analyzes to support a mono-dimensional structure. Also, 24-total item for a tool with two dimensions which are so high correlated maybe can be reduced. Perhaps, rather than conducting two separate confirmatory analyzes for the two instruments, it could be useful to do a single CFA on the 24 items to see if the two-factor structure is confirmed... Did the author so any considerations about this? If the authors choose to maintain the two independent tools, maybe additional theoretical justifications for such a choice would be appropriate.

As we also noted in the “Discussion” section, we agree that the lack of discrimination between the two subscales is the weakest aspect of the paper. In this respect, we have added alternative models based on the CFA to both the results and the conclusion. Beyond what is offered in the paper, we would like to explain our position on this. Despite the lack of discrimination found in the analyses, we believe that given that the fit of the models is adequate, it is important to give weight to other types of validity evidence, such as content-based validity evidence. We also believe that it is important not to modify the scale a priori, since the items included are representative and relevant to the construct we want to assess. Moreover, although the subscales are used jointly in this study, some authors may choose to use only one of the two subscales. In other words, we believe that although, as we have seen, they can be used as a unidimensional scale, they can also be used as independent scores, always recognizing their potential for discrimination if they are included in the same analysis.

  1. I would recommend the authors to deepen the limitations section of the manuscript. The authors only cite the fact that that only nine participants had trans family members. There are other aspects that the authors might consider (the generalizability of the results, the high correlations of the two tools, etc… -some aspects are reported in the conclusions-)

We have extended the limitations accordingly.

  1. I appreciated the conclusions section. I would stress more with a couple of sentences the relevance to represent more the ‘T’ and the ‘Q’ people of the LGBTQ acronym, in the scientific research too. This paper contributes to this, and such a tool might be useful for other researchers who want to do research on LGBTQ+ issues, making trans people less invisible and ignored. Maybe the authors might read and cite the following reference to elaborate more on this point: Salvati, M., & Koc, Y. (2022). Advancing research into the social psychology of sexual orientations and gender identities: Current research and future directions. European Journal of Social Psychology52(2), 225-232. https://doi.org/10.1002/ejsp.2875

We have included the reference and expanded the conclusions.

Round 2

Reviewer 2 Report

I have read the new version of the manuscript and the detailed responses of the authors to my previous comments. I think they did a great job and have responded to my revision requests satisfactorily. I found the manuscript improved and I'm glad to recommend it for publication on IJERP.

Best regards,